# The impact of the exercise on the social mentality of the Chinese people

Shuyu Ji[1☯], Kaiqi Zhang[2☯], Ludan Xu[3], Xiaolin Wang[4‡]*, Delong Dong[2‡]*, Xiannan Yang[1‡]*

1 College of Physical Educational and Sports, Beijing Normal University, Beijing, China, 2 Physical Education, Ludong University, Yantai, Shandong, China, 3 College of Foreign Languages, Zhejiang University of Finance and Economics, Hangzhou, Zhejiang, China, 4 Faculty of Educational Studies, Department of Sport Studies, University Putra Malaysia, Subang Jaya, Malaysia

☯ These authors contributed equally to this work.
‡ These authors also contributed equally to this work
* GS64854@student.upm.edu.my (XW); dongdelong@ldu.edu.cn (DD); yangxiannan@bnu.edu.cn (XY)

## Abstract

### Objective

Engaging in exercise not only encompasses its intrinsic attributes but also signifies its social dimensions. It reflects an underlying emotional tone and cumulative value attributed to exercise by individuals, forming a broad, macro-level socio-psychological relationship in alignment with the conceptual definition of societal mentality. The social mentality is an indicator of a nation's governance capacity and mirrors the overall socio-psychological profile and needs of its citizens. This study, therefore, aims to investigate the influence of exercise, specifically through the lens of sports, on the social mentality of the Chinese population, encompassing aspects of social trust and social equity. Additionally, we explore the distinct mechanisms underlying differences in social class, generation disparities, and spatial dimensions. This inquiry aims to contribute to enhancing governance capabilities and societal stability.

### Method

Data for this study was sourced from the *2023 China General Social Survey*. We selected 20 variables and analyzed a sample of 6,746 individuals. We employed Ordinary Least Squares (OLS) multiple linear regression models to construct our analysis.

### Results

The findings indicate that exercise has a positive influence on the social mentality of the Chinese population. A higher frequency of participation in sports correlates with a more advanced level of social mentality development. Notably, significant disparities exist between the macro-level and micro-level impacts of exercise, suggesting ample room for improvement. Social class, generational disparities, and spatial dimensions demonstrate substantial impact, each exhibiting unique characteristics depending on the specific

**Data Availability Statement:** All relevant data are within the manuscript and its Supporting Information files.

**Funding:** The author(s) received no specific funding for this work.

**Competing interests:** The authors have declared that no competing interests exist.

research question. Furthermore, the weightings of social trust orientation and social equity orientation within the societal mentality dimensions exhibit variability and fluctuations.

## Conclusion

When exploring the topic of social mentality, it is recommended to separately discuss the various dimensions it encompasses, thus providing a comprehensive, detailed, and precise portrayal of specific issues. To bolster the influence of exercise on the social mentality, as well as to enhance governance capabilities and societal stability, the following recommendations are proposed: (1) In-depth exploration of differences within social strata to optimize the mechanisms through which exercise influences the social mentality; (2) Balancing generational disparities to establish a solid foundation for the influence of exercise on the social mentality; (3) Recognizing spatial dimensions to harness the spatial dynamism of exercise in shaping the social mentality.

## Introduction

Movement has not only natural properties [1,2], but also social properties [3]. Increasing productivity has led to a shift towards diversification of the material needs of the Chinese people [4]. In particular, the increasing demand for a long and healthy life has stimulated the Chinese people's interest and passion for sports. Under conditions of a proliferating exercise population, there has been a gradual shift from the initial use of the natural attributes of exercise for the purpose of achieving a long and healthy life to a reliance on the social attributes. The main expression is the sum of emotional tone, social consensus, and value orientation, which is a broad, macroscopic, socio-psychological relationship between individuals and groups [5]. That is, the Chinese people's participation in the movement presents a unique social mentality. The characteristics and changing movements of social mentality reflect the overall style of different times and different regions, and are an important way of presenting the will of the people [6]. The current academic research on social mentality is generally characterized by two views: positive and negative. Positive view:① Positive social mindset is an important factor in regulating and overcoming loneliness and optimizing the impact of the resource environment on mental health through a closed-loop dynamic linkage of stress-mindset-exercise-feedback [7]. ② Positive social mindset guides the consumer value orientation dominated by social comparison and personal will, with a greater probability of avoiding negative emotions and economic losses triggered by coercive transactions, safeguarding consumers' rights and interests and increasing positive emotions.③ Positive social mindset significantly predicts ethnic identity, cultural adaptation, and social support [8], contributes to the enhancement benefits of positive social mindset on people's physical and mental health through the mediating factors of social integration [9], promotes social connectedness, and relieves psychological stress [10]. Negative point of view:① Negative social mentality is prone to lead to network addiction, although there are differences in the situation of different age stages, but the overall presentation of the people's social desires are falsely satisfied, leading to bias in social cognition, which in turn eliminates the positive social mentality, forming a vicious circle of negative social mentality and social cognitive bias [10,11]. ② Negative social mentality leads to a crisis of confidence in the policies proposed by the government, especially the skepticism of the lower class people is even stronger, which seriously affects the implementation of policies and the green

transformation of social ecology [12]. ③ Negative social mentality severely inhibits sustainable urban development and people's sense of well-being, while at the same time, the decline of mental health leads to the violation of physical health, and the extent of this inhibition and violation can be superimposed by major security risk events [13,14]. For the exercise side of things, the relevant studies point out: Sports can shape the collective consensus, can provide a psychological discipline for individuals to participate in sports, so that individuals in the "circle society" to build "self-ethics", leading to a positive social mentality [15]. Therefore, the impact of the movement on the social mentality of the Chinese people deserves to be explored in depth.

Previous studies have pointed out that people's social mentality is influenced by multiple inequalities of class, generation, and space. Class inequality aspects:① Government trust support and government trust crisis present a differentiated development of class inequality, with social mentality status as a key mediating factor. That is, the upper and middle classes generally have a positive social mentality and show trust and support for the government, while the lower classes are the opposite [12]. ② Social mentality is a central element of the power and dominance relationship between different classes, and constructivist theoretical studies have pointed out that the upper class positions the social mentality of the middle class over moral superiority, and the social mentality of the lower class over the double plunder of moral superiority and material deprivation in order to realize the power and dominance relationship between the classes [16]. ③ There is a situation of class-service disparity between favored groups and discriminated targets in the credit operations of financial institutions, which are mainly oriented towards the upper and middle classes and generally discriminate against the lower classes. This exacerbates the "social mentality of the underprivileged" of the lower classes, which, under the influence of higher interest rates, creates a vicious circle of negative social mentality [17]. ④ Whether people's sporting needs are met or not, there is a facilitating or inhibiting relationship on the social class of their families and individuals, which directly affects the direction of positive or negative development of social mentality [18]. At the same time, organized movements can alleviate class inequality and economic capital plays an important role and is an essential element in reducing class disparities through movements [19]. Aspects of intergenerational inequality:① Differences in exercise across generational cohorts influenced expected class goals, with generational cohorts that consistently maintained exercise participation having a higher class in adulthood than expected in childhood, i.e., exercise participation facilitated upward mobility [20]. ② Observed longitudinally over the life course, exercise reduces health inequalities across generational groups, particularly in cardiovascular disease, and specific exercise prescriptions and health risk prevention screening measures should be set by generational group characteristics to prolong the human lifespan and to avoid negative social mindsets due to disease [21]. Spatial inequality aspects:① Southern versus northern differences in social welfare have led to significant heterogeneity in the social mentality of the Brazilian people [22], and to a southern versus northern heterogeneity in the health inequalities of the Finnish people. At the same time, urban-rural spatial differences in health inequalities have triggered a continuous trend of expansion of negative social mentality [23]. ② Corporate philanthropy between China and the U.S., as well as between regions within China, exhibits spatial inequality issues in terms of practitioners' information perceptions, information availability, and corporate operational effectiveness and financial pressures, leading to a differentiated social mindset among the people due to spatial inequality in philanthropy [24]. ③ The spatial inequality of household consumption status triggers the differentiated social mentality of Chinese people, especially in energy consumption, which is much higher in the central region than in the east and west. At the same time, the social mentality of the people in the central region is skewed towards the negative, while the eastern and western regions are skewed towards the positive [25].

In view of this, the Chinese General Social Survey (CGSS) released in 2023 was used as the data source for this study. OLS multiple linear regression models were constructed. Considering considerations of class, generation, and space, it delves into the impact of the movement on the social mentality of the Chinese people. To provide a reference basis and development inspiration for enhancing the social mentality of the Chinese people through sports, and to promote the quality and quantity of China's human development level.

## Study design

### Data sources, sample selection, indicator selection

1. **Data sources:** The data for this study were derived from the Chinese General Social Survey (CGSS). The project is hosted by Renmin University of China and the China Survey and Data Center. Comprehensive and systematic collection of multi-level data on society, communities, families, and individuals. Trends in social change are summarized and topics of great scientific and practical significance are explored. Promote the opening and sharing of domestic resources and support international comparative research. Launching the East Asian Social Survey (EASS) with the aim of providing reliable data support for a wide range of researchers.

2. **Sample Selection:** Given the reliability of the program, its data release in 2023 was selected for this study. The total sample size for this data was 8,148 and the total number of variables was 700. After excluding "don't know", "can't answer" and missing values, the remaining valid sample was 6746. After eliminating the null and non-target variables, the remaining target variables were 20.

3. **Indicator selection:**① Social mentality (dependent variable): The relevant viewpoints of academic research on social mentality have been described above [6–15], therefore, taking into account the indicators used in existing studies, social trust and social fairness are used to reflect the social mentality of the Chinese people.② Exercise (impact core): Walking, running, physical training, square dancing, ball games and other programs are used as the main activities to move the body to the state of sweating, and the form is free to choose, single, group, and combined can be. The reflective indicator is the frequency of exercise under the combined above conditions.③ Class (independent variable): the current academic community possesses a variety of class theories and divisions [12,16–19]. For sports, Bourdieu constructed the theory of social class in sports, which is divided by economic capital, social capital, and cultural capital [26], and the theory and division are widely used in the study of sports and social issues. Among the indicators reflecting economic capital are housing size, home ownership, and income level [26]; Reflective indicators of social capital are economic status, work status [27]; Cultural capital is reflected by indicators of cultural activities, education, and frequency of socialization [28]. ④ Generation (dependent variable): for the division of generations. Chinese scholars mainly adopt 2 approaches: One is to divide them according to a certain point in time, such as 1988 as the time boundary, before the birth of the old generation, after the birth of the new generation [29]; The second is to use 10 years as a criterion for delineation, and those born during each 10-year period are considered as one generation, such as 60th, 70th, 80th, etc. [30]. International scholars mainly use a 4-generation division: Traditional Generation(Born before 1946 or before World War II. Generation "T" for short)、Baby Boomers(Born 1946–1964. abbreviated "B" generation)、Generation Xers(Born 1965–1980. referred to as Generation "X")、Generation Years(Born in 1981 and later. Abbreviation "Y" generation)。This is categorized as a

"major socio-historical event". As this study aims to provide reference examples and templates for countries around the world on how the movement can improve people's social mentality and enhance national governance capacity. Therefore, a 4-generation division was chosen as the basis for the generational division in this paper. That is, Generation "T", Generation "B", Generation "X" and Generation "Y".⑤ Space (dependent variable): The problems of social mentality caused by spatial inequality have been described above [22–25]. For this reason, based on the characteristics of the Chinese people and the geographic features of China, the classic 3 major geographic divisions of China were chosen as the spatial division criteria. That is, eastern, central, and western.⑥ Control variables: According to related studies in academia [12,26–28], gender, marriage, household registration, health insurance, pension insurance, physical health, and mental health were used as control variables. Generations were included as control variables when sub generational discussions were not conducted.

## Model setup

Reflects the social mentality of the Chinese people through 2 specific dimensions: social trust and social equity. Pinar [31] (2013) states that when common problems or common indicators are reflected through multivariate, multidimensional, or multilevel. There is often a disparity in weighting between multiple variables, dimensions, or levels. They should be counted separately. The study by Qiu [32] (2018) pointed out that when there are multiple variables reflecting common indicators in the study. If the nature of the problem is to be explored in detail and in depth. It should be discussed separately rather than analyzed uniformly after weighted fitting. Therefore, this study draws on the research results of the above scholars. The dependent variable (Chinese people's social mentality) was refined into two specific measures, namely, social trust and social fairness.

OLS multiple linear regression models were used. The relationship of the variables in this study is a relationship between a dependent variable and multiple independent variables, therefore, the OLS multiple linear regression model is more applicable [33]. However, considering that the OLS multiple linear regression model must exclude the problem of multicollinearity between the dependent and independent variables, a diagnosis of multicollinearity is performed. The results are shown in Table 1 and the VIF values are all less than 5, indicating that there is no problem of multicollinearity.

After excluding multicollinearity. The constructed regression model equation is as follows:

$$Y_i = X_i + \varepsilon_i \tag{1}$$

This represents Model 1 (baseline) used to examine the social mentality of the Chinese people under normal circumstances. It provides a reference for comparisons with subsequent models that incorporate additional relevant factors.

$Y_i$ represents the social mentality of the Chinese people (dependent variable). The subscript $i$ stands for the two dimensions of social mentality: social trust and social equity. $X_i$ corresponds to the control variables which include demographic factors and social security factors. When intergenerational variable classification is not discussed, they are included within the demographic variables. $\varepsilon_i$ represents the constant term.

$$Y_i = LnOSport_i + X_i + \varepsilon_i \tag{2}$$

This is Model 2 (exercise), which aims to examine whether exercise significantly influences the social mentality of the Chinese people. $LnOSport_i$ represents the exercise variable that

**Table 1. Diagnosis of multicollinearity.**

|  | Social trust | | Social equity | |
|---|---|---|---|---|
|  | **Tolerances** | **VIF** | **Tolerances** | **VIF** |
| Exercise | .853 | 1.172 | .853 | 1.172 |
| Gender | .877 | 1.140 | .877 | 1.140 |
| Generation | .573 | 1.745 | .573 | 1.745 |
| Marriage | .901 | 1.110 | .901 | 1.110 |
| Household registration | .731 | 1.368 | .731 | 1.368 |
| Medical insurance | .941 | 1.063 | .941 | 1.063 |
| Pension | .846 | 1.182 | .846 | 1.182 |
| Physical Health | .699 | 1.430 | .699 | 1.430 |
| Mental health | .795 | 1.257 | .795 | 1.257 |
| Housing size | .990 | 1.010 | .990 | 1.010 |
| Homeownership | .867 | 1.154 | .867 | 1.154 |
| Income levels | .734 | 1.363 | .743 | 1.363 |
| Economic status | .883 | 1.133 | .883 | 1.133 |
| Employment status | .763 | 1.311 | .763 | 1.311 |
| Cultural activities | .879 | 1.137 | .879 | 1.137 |
| Educational level | .548 | 1.826 | .548 | 1.826 |
| Social frequency | .957 | 1.045 | .957 | 1.045 |
| Geographical subdivision | .887 | 1.127 | .887 | 1.127 |

Note: Owing to space constraints in the table, when the first digit is "0", the "0" is omitted and is used below.

affects social mentality.

$$Y_i = LnOSport_i + InClass_{i,j} + LnOSport_i \times InGeneration_{i,k} + LnOSport_i \times InSpatial_{i,f} + X_i + \varepsilon_i \quad (3)$$

This is Model 3 (class), aims to investigate the influence of exercise on the social mentality of the Chinese people. Additionally, it examines the interactive effects of exercise with generations and spatial factors, considering the combined effects of social class and control variables. $InClass_{i,j}$ represents social class, where $j$ signifies the dimensions within the social class. $InGeneration_{i,k}$ is the categorical variable accounting for intergenerational differences, where $k$ denotes four distinct generational groups: TG, BB, GX, and GY. $InSpatial_{i,f}$ serves as a categorical variable considering spatial differences, with $f$ indicating the Eastern, Central, and Western regions. Building upon research by Korkmaz (2021) and Davison (2021), both of which underscore the importance of testing interactions between the core dependent variable and variables subject to categorization, particularly in the context of OLS multiple linear regression analysis, we have adopted a similar approach. This ensures that the status of the core dependent variable remains consistent in subsequent statistical analyses after categorization. In Model 3 (class), we initially investigate the interactions between exercise and generations, followed by the examination of interactions between exercise and spatial factors. Subsequently, while categorically discussing generations and spatial factors, the exercise variable as the core dependent variable was maintained to ensure consistency. Therefore, the equation, $LnOSport_i \times InGeneration_{i,k}$ represents the interaction term between exercise and generations, and $LnOSport_i \times InSpatial_{i,f}$ represents the interaction term between exercise and spatial factors. As this section necessitates the validation of the aforementioned interactions and scrutinizes the influence of exercise on the societal mentality of the Chinese population, particularly

within the context of social class, the exercise variable is retained in both the equation and the presentation of statistical results. However, the exercise variable was no longer presented separately during the analysis of interaction effects.

Since human class identity is objective.Therefore, when engaging in a sub-generational and spatial discussion. The variables included in social class are incorporated. In order to explore the impact of the movement on the social mentality of the Chinese people in a pluralistic, comprehensive, and real way. At the same time, as the sub-generational and spatial statistics are conducted. The generational and spatial variables were turned into categorical variables and could no longer be set up as interaction terms with the movement variables.Moreover, the interaction tests between movement and generation, and movement and space have already been conducted in Model 3 (strata) above. Therefore, the interaction terms are not set in the generational and spatial statistics. The generational variables were categorized as categorical variables, and model 4 ("T" generation), model 5 ("B" generation), model 6 ("X" generation), and model 7 ("Y" generation) were constructed. "Y" generation) are formulated as follows:

$$Y_{i,k} = LnOSport_{i,k} + InClass_{i,j,k} + X_{i,k} + \varepsilon_{i,k} \tag{4}$$

Constructing Model 8 (Eastern area), Model 9 (Middle area), and Model 10 (Western area) with spatial variables as categorical variables, the equation is as follows:

$$Y_{i,f} = LnOSport_{i,f} + InClass_{i,j,f} + X_{i,f} + \varepsilon_{i,f} \tag{5}$$

## Variable manipulation and description

**Dependent variable (social mentality of the Chinese people):** Social trust: Among the CGSS questionnaires, the survey on social trust was designed to ask the Chinese people about their attitudes toward social trust, positively coded. Social equity: Among the questionnaires in the CGSS, the survey on social equity was designed to ask the Chinese people about their attitudes toward social equity, positively coded.

**Influence on core variables (exercise):** The exercise variable is the core variable of influence in this paper. The frequency distribution of the reflective indicators is "daily = 1, weekly = 2, monthly = 3, yearly = 4 and never = 5", reverse coded.

**class (dependent variable):** Housing area is taken as its original value excluding invalid responses, positively coded. Housing equity is dichotomized and positively coded for presence or absence. Income levels are specific values in the CGSS. Considering the basic situation of people's income in China's social formation and referring to scholars' research literature [26]. Segmentation of income levels to be more in line with the realities of Chinese society and positive coding. The economic status is reverse coded in accordance with the CGSS. Work status is based on the situation at the time of the survey, and similar situations such as "yes before, not now" are included in "no work", which is reverse coded. Cultural activities are reverse coded in accordance with the CGSS. Education was coded forward from low to high. Socialization frequency was reverse coded according to the operationalization of the CGSS.

**Generations (dependent variable):** Generation "T" (born before 1946) = 1, Generation "B" (born between 1946–1964) = 2, Generation "X" (born between 1965–1980) = 3, and Generation "Y" (born after 1981) = 4, coding is predominantly positive and categorical. 1980) = 3, "Y" generation (born after 1981) = 4, coding is predominantly positive and categorical.

**Space (dependent variable):** "Eastern region" (including Beijing, Tianjin, Hebei, Liaoning, Shanghai, Jiangsu, Zhejiang, Fujian, Shandong, Guangdong, Hainan) = 1; "Central region" (comprising Shanxi, Jilin, Heilongjiang, Hubei, Hunan, Henan, Anhui, Jiangxi) = 2; "Western region" (Inner Mongolia, Chongqing, Sichuan, Guizhou, Yunnan, Tibet, Shaanxi, Gansu,

**Table 2. Descriptive statistics of the sample.**

| Variables | Variables | Coding Method | M (SD) | Attributes |
|---|---|---|---|---|
| Dependent variables | Social trust | Continuous Variable with Values 1–5 | 3.657(.988) | + |
| | Social equality | Continuous Variable with Values 1–5 | 3.467 (.959) | + |
| Independent variable | Exercise | Every day = 1; Several times a week = 2; Several times a month = 3; Several times a year = 4; Never = 5 | 3.128 (1.608) | – |
| Demographic Characteristics | Gender | Male = 1; Female = 2 | 1.527 (.499) | Categorized |
| | Generation | TG = 1; BB = 2; GX = 3; GY = 4 | 2.769 (.929) | +/ Categorized |
| | Marriage | Single = 1; Married = 2 | 1.733 (.441) | Categorized |
| | Household registration | Rural = 1; Urban = 2 | 1.431 (.495) | Categorized |
| Social security | Medical insurance | Yes = 1; No = 2 | 1.055 (.228) | – |
| | Pension | Yes = 1; No = 2 | 1.261 (.439) | – |
| | Physical health | Continuous Variable with Values 1–5 | 3.495 (1.074) | + |
| | Mental health | Continuous Variable with Values 1–5 | 3.972 (1.063) | + |
| Class | Housing size | Continuous Variable | 122.377 (180.868) | + |
| | Homeownership | Yes = 1; No = 2 | .503 (.500) | + |
| | Income levels | Below 30000 RMB = 1; 30000–60000 RMB = 2; Above 60000 RMB = 3 | 1.775 (.824) | + |
| | Economic status | High level = 1; Upper-middle level = 2; Middle level = 3; Lower-middle level = 4; Lower level = 5 | 3.706 (.894) | – |
| | Employment status | Yes = 1; No = 2 | 1.477 (.499) | – |
| | Cultural activities | Every day = 1; Several times a week = 2; Several times a month = 3; Several times a year = 4; Never = 5 | 4.516 (.761) | – |
| | Educational level | No education = 1; Junior high school or below = 2; High school / vocational school = 3; Junior college / bachelor's degree = 4; Master's degree or above = 5 | 2.537 (.954) | + |
| | Social frequency | Every day = 1; Several times a week = 2; Several times a month = 3; Several times a year = 4; Never = 5 | 2.646 (1.109) | – |
| Space | Geographical subdivision | Eastern area = 1; Central area = 2; Western area = 3 | 1.850 (.813) | +/ Categorized |

Qinghai, Ningxia, Xinjiang, Guangxi) = 3. The geographic division codes in Table 1 are shown as abbreviations for the provinces.

**Control variables:** Gender is only categorized as male or female. Age is operationalized in the section on intergenerational differences. Marriage is coded as single and married. Household registration is coded as "current family household and former agricultural (rural) household under the agricultural (rural) household type; current family household and former non-agricultural (urban) household under the non-agricultural (urban) household type". Medical insurance and pension insurance were coded with or without dichotomization. Physical health and mental health were coded as positive. Descriptive statistics for each sample variable are shown in Table 2.

## Statistical results

### The impact of the movement under class conditions on the social mentality of the Chinese people

The direction of influence and significance of the control variables are the same for Models 1 and 2. As can be seen from Table 3, in Models 1 and 2, the social mentality of male residents is better than that of females; the social mentality of the four generational groups "T, B, X, Y" is

**Table 3. Analysis of class differences in the movement's impact on the social mentality of the Chinese people.**

| Variables | Model 1 (basic) | | Model 2 (exercise) | | Model 3 (class) | |
|---|---|---|---|---|---|---|
| | ST | SF | ST | SF | ST | SF |
| Gender | -.073(.023)** | -.066(.023)** | -.071(.023)** | -.065(.023)** | -.070(.025)** | -.083(.024)** |
| Generation | -.151(.013)*** | -.128(.013)*** | -.153(.013)*** | -.129(.013)*** | \ | \ |
| Marriage | .003(.027) | -.016(.026) | .005(.027) | -.015(.026) | .025(.028) | -.009(.026) |
| Household Registration | .001(.024) | -.012(.023) | -.018(.024) | -.023(.024) | -.022(.027) | -.032(.026) |
| Medical insurance | -.142(.052)** | -.135(.051)** | -.139(.052)** | -.133(.051)** | -.112(.053)* | -.096(.050)+ |
| Pension | -.003(.028) | .028(.027) | -.001(.028) | .029(.027) | -.004(.028) | .029(.027) |
| Physical health | .031(.012)* | .067(.012)*** | .027(.012)* | .065(.012)*** | .007(.013) | .041(.012)** |
| Mental health | .097(.012)*** | .111(.011)*** | .096(.012)*** | .111(.011)*** | .097(.012)*** | .104(.011)*** |
| Exercise | | | -.025(.007)** | -.015(.007)* | .076(.015)*** | .040(.015)** |
| Exercise × Generation | | | | | -.042(.004)*** | -.033(.004)*** |
| Exercise × space | | | | | .007(.004)+ | .020(.004)*** |
| Housing size | | | | | -.001(.001) | -.001(.001) |
| Homeownership | | | | | .033(.025) | -.011(.024) |
| Income levels | | | | | -.039(.016)* | -.008(.016) |
| Economic status | | | | | -.066(.014)*** | -.157(.013)*** |
| Employment status | | | | | -.033(.026) | -.008(.025) |
| Cultural activities | | | | | -.016(.016) | -.026(.015)+ |
| Educational level | | | | | .027(.016)+ | -.001(.015) |
| Social frequency | | | | | .037(.010)** | .004(.010) |
| Constant | 3.625(.064)*** | 3.194(.062)*** | 3.731(.071)*** | 3.257(.069)*** | 3.621(.143)*** | 3.747(.137)*** |
| N | 6746 | 6746 | 6746 | 6746 | 6746 | 6746 |
| $R^2$ | 0.034 | 0.038 | 0.036 | 0.039 | 0.041 | 0.061 |
| Adj $R^2$ | 0.033 | 0.037 | 0.035 | 0.038 | 0.039 | 0.058 |

Note: ST = social trust, SF = social fairness. + = $P<0.1$、

* = $P<0.05$、

** = $P<0.01$、

*** = $P<0.001$.Standard errors in parentheses.

gradually weakened in accordance with the order of the social mentality; and there is no significant effect of marriage, household registration and pension insurance; Those who have medical insurance have better social mindfulness; those who are physically and psychologically healthier have better social mindfulness benefits. Model 2 shows that sports can significantly influence the social mindset of the Chinese people. The higher the frequency of participation in sports, the better the social mindset.

In Model 3, the introduction of the class and interaction terms leads to fluctuations in the control variables. Medicare's ability to influence has decreased, and the significance for social equity has fallen between 0.05 and 0.1, a large decline. The ability to influence physical health is reduced, and trust in society has been reduced to non-significant. The age (generational) variable was not presented in this section due to the need for interaction terms. Gender, marriage, household registration, pension insurance, and mental health are the same as in Models I and II, as described above. Among the class variables, housing size, home ownership, and work status failed to have a significant effect. Income level only significantly affects social trust, in the sense that the lower class outperforms the middle class and then the upper class and has no effect on social equity. Economic status can produce significant benefits in terms of the campaign's impact on the social mindset of the Chinese people. Expressed as a superiority

effect from upper to lower levels. Cultural activities can have a weakly significant effect on social equity, manifested as a higher frequency, a more favorable social mindset, and no effect on social trust. Educational attainment and socialization frequency had a significant effect on social trust, as indicated by a weakly significant high to low dominance effect for educational attainment, and a more significant low to high dominance effect for socialization frequency, with no effect on social equity. Observing from the value of $R^2$, Social equity ($R^2 = 0.061$) was more developed than social trust ($R^2 = 0.041$). The interaction terms for movement and generation and movement and space were significant. It shows that campaigns can have a significant impact both generationally and spatially. Therefore, when discussed spatially and generationally, movement can still serve as the core of influence.

## Endogeneity and robustness tests

**(1) Endogeneity test.** To avoid potential reverse causation, which would lead to the results of the study above being confounded by endogeneity issues. This paper utilizes instrumental variables approach to reduce this endogeneity problem. This paper replaces the core explanatory variables with whether they exercise. Following Qiu's [32] (2018) guidance on model construction, the movement of the core explanatory variable and whether the instrumental variable is moving constitute an interaction term to be tested. The results in Table 4 show that the replaced instrumental variables remain consistent with the results above.

**(2) Robustness check.** In order to prove the reliability of the statistical results of this study, 2 ways of testing were taken:① Replacing the dependent variable. Summing up social trust and social equity, with a single social mindset as the dependent variable.② Propensity score matching. The total sample was divided into experimental and control groups according to the mean values of the exercise variables, using the control variables from above as matching variables. As shown in Table 5, the movement still has a significant contribution to the social mentality of the Chinese people.

## The impact of the movement on the social mentality of the Chinese people under generational differences

The statistical results according to the different generational groups are shown in Table 6. Model 3 above is a macro-view observation that the movement has been shown to contribute to the social mentality of the Chinese people under various types of influencing factors. In contrast, the micro-observations in Table 6 are sub-generational, and the campaign produces a lower impact potency, with only a weakly significant boost to social trust in Generation Y.

Model 4 is for the population born before 1946. Gender, marriage, pension insurance, physical health, housing ownership, income level, and work status failed to significantly influence the social mentality of this group. Household registration has a significant negative effect on

**Table 4. Endogeneity test.**

| | ST | | | ST | | |
|---|---|---|---|---|---|---|
| | (1) Exercise | (2)ST | (3)ST | (1) Exercise | (2)SF | (3)SF |
| Exercise or not | 2.859(.016)*** | -.018(.009)* | | 2.859(.016)*** | -.024(.009)** | |
| Exercise ×Exercise or not | | | -.006(.003)** | | | -.008(.003)** |
| Control variable | Yes | Yes | Yes | Yes | Yes | Yes |
| Constant | -.758(.033)*** | 3.816(.030)*** | 3.688(.019)*** | -.758(.033)*** | 3.410(.029)*** | 3.509(.018)*** |
| N | 6746 | 6746 | 6746 | 6746 | 6746 | 6746 |
| $R^2$ | 0.694 | 0.034 | 0.001 | 0.694 | 0.038 | 0.001 |

**Table 5. Robustness test.**

| | Social mentality | ST | | SF | |
|---|---|---|---|---|---|
| | | Test group | Control subjects | Test group | Control subjects |
| Exercise | -.041(.012)*** | -.054(.020)** | -.011(.043)* | -.026(.019)* | -.117(.042)** |
| Control variable | Yes | Yes | Yes | Yes | Yes |
| Constant | 7.215(.186)*** | 3.703(.158)*** | 3.943(.272)*** | 3.348(.153)*** | 4.109(.263)*** |
| N | 6746 | 3705 | 3041 | 3705 | 3041 |
| $R^2$ | 0.053 | 0.042 | 0.032 | 0.043 | 0.039 |

social trust. Manifestation of social trust in the rural population is more advantageous than in the urban population. No effect on social equity. Health insurance has a significant negative impact on social equity. Demonstrates that having health insurance is more conducive to the development of social equity. No effect on social trust. Mental health has a positive and significant impact on the social mentality of the group. This is reflected in the fact that the higher the level of mental health, the higher the level of development of social mentality. Housing size has only a weakly significant effect on social trust and no effect on social equity. Economic status has a significant negative impact on social equity. Expressed as a dominance effect from the top to the bottom. No effect on social trust. Both cultural activities and education have a significant positive effect on social trust. The lower the level of participation in cultural activities and the higher the level of education, the higher the level of social trust development. Socialization frequency has only a weakly significant negative effect on social equity. Showed that higher frequency of socialization was associated with higher levels of development of social equity, but the effect was limited. There is no effect on social trust. From the value of $R^2$ it is observed that social trust ($R^2 = 0.085$) is better developed than social equity ($R^2 = 0.067$).

**Table 6. Analysis of the impact of sports on the social mentality of the Chinese people under generational differences.**

| Variables | Model 4 (TG) | | Model 5 (BB) | | Model 6 (GX) | | Model 7 (GY) | |
|---|---|---|---|---|---|---|---|---|
| | ST | SF | ST | SF | ST | SF | ST | SF |
| Gender | .098(.090) | .058(.091) | -.077(.044)+ | -.161(.043)** | -.039(.048) | -.099(.046)* | -.087(.045)+ | -.013(.042) |
| Marriage | -.043(.086) | -.045(.087) | .023(.050) | -.031(.050) | .136(.067)* | .089(.064) | .067(.049) | .054(.046) |
| Household Registration | -.241(.109)* | -.145(.109) | -.075(.051) | -.074(.050) | .010(.054) | .018(.052) | .049(.046) | -.046(.043) |
| Medical insurance | -.076(.167) | -.347(.167)* | -.008(.092) | -.133(.091) | -.046(.107) | -.125(.101) | -.272(.090)** | .006(.084) |
| Pension | -.134(.105) | .087(.106) | -.031(.054) | -.033(.053) | -.026(.058) | .048(.055) | .036(.048) | .041(.044) |
| Physical health | -.063(.042) | .069(.043) | .011(.020) | .037(.020)+ | -.017(.024) | -.002(.023) | .072(.027)** | .101(.025)*** |
| Mental health | .168(.042)** | .103(.042)* | .062(.020)** | .067(.020)** | .100(.023)*** | .137(.022)*** | .118(.024)*** | .103(.022)*** |
| Exercise | -.035(.025) | -.021(.025) | -.018(.012) | -.007(.012) | -.007(.015) | -.002(.014) | -.029(.017)+ | -.010(.016) |
| Housing size | -.001(.001)+ | .001(.001) | .001(.001) | .001(.001) | .001(.001) | -.001(.001) | -.001(.001)+ | -.001(.001) |
| Homeownership | -.086(.086) | -.113(.086) | .014(.043) | -.070(.042) | .122(.047)** | .081(.044)+ | -.013(.050) | .019(.046) |
| Income levels | -.103(.066) | -.044(.067) | -.008(.032) | -.026(.031) | -.017(.031) | -.024(.029) | -.046(.028) | .014(.026) |
| Economic status | -.035(.046) | -.095(.046)* | -.075(.022)** | -.179(.022)** | -.089(.027)** | -.179(.025)** | -.006(.028) | -.078(.026)** |
| Employment status | .129(.149) | -.137(.150) | -.076(.046)+ | -.014(.045) | -.149(.052)** | -.107(.049)* | .033(.053) | .059(.049) |
| Cultural activities | .105(.053)* | .033(.054) | -.013(.026) | -.022(.026) | -.059(.032)+ | -.073(.031)* | -.022(.031) | -.009(.029) |
| Educational level | .207(.064)** | .051(.064) | -.025(.032) | -.077(.031)* | .024(.034) | -.009(.032) | .050(.027)+ | .042(.025)+ |
| Social frequency | .001(.033) | -.062(.033)+ | .023(.016) | -.014(.016) | .048(.021)* | .045(.020)* | .058(.022)* | .016(.021) |
| Constant | 3.030(.527) | 3.993(.530)** | 4.023(.231)** | 4.362(.227)** | 3.707(.279)** | 3.911(.264)** | 2.690(.286)** | 2.616(.267)** |
| N | 480 | 480 | 2417 | 2417 | 2012 | 2012 | 1837 | 1837 |
| $R^2$ | 0.085 | 0.067 | 0.023 | 0.055 | 0.049 | 0.084 | 0.047 | 0.041 |

Model 5 is for the population born between 1946 and 1964. Marriage, household registration, health insurance, pension insurance, housing size, home ownership, income level, cultural activities and frequency of socialization failed to have a significant impact on the social mentality of this group. Gender has a significant negative impact on social mentality. It is manifested in the fact that males have a better level of development than females. In this case, it is weakly significant for social trust. Physical health has only a positive and weakly significant effect on social equity. It is shown that higher levels of physical health are associated with higher levels of social equity development. There is no effect on social trust. Mental health has a positive and significant effect on social mindfulness. This is manifested in the fact that the higher the level of mental health, the higher the level of social mentality. Economic status has a negative and significant effect on social mentality. This is characterized by a dominance effect from the top to the bottom of the hierarchy. Work status has only a negative and weakly significant effect on social trust. Expressed as a higher level of social trust development with a worker. There is no effect on social equity. Educational attainment has a significant negative effect on social equity. Showed higher social equity development for those with less education. There is no effect on social trust. From the value of $R^2$, it is observed that social equity ($R^2 = 0.055$) is better developed than social trust ($R^2 = 0.023$).

Model 6 is for the population born between 1965 and 1980. Household registration, health insurance, pension insurance, physical health, housing size, income level and education failed to significantly influence the social mentality of this group. Gender had a significant negative effect on social equity. Manifested in a better level of development for males than for females. No effect on social trust. Marriage has a significant positive impact on social trust. It is shown that married people have a better level of development than single people. No impact on social equity. Mental health has a significant positive impact on social mindfulness. This is reflected in the fact that higher levels of mental health are associated with higher levels of social mindfulness. Home ownership has a significant positive impact on social mentality. This is manifested in the fact that those who own home ownership have a better level of development than those who do not own home ownership. In this case, it is weakly significant on social equity. Economic status, work status and cultural activities can have a significant negative effect on social mentality. This is reflected in the dominant effect of economic status from the upper to the lower strata, the higher level of social mentality of those who have a job, and the higher level of social mentality of those who participate in cultural activities more frequently. Among them, cultural activities are weakly significant on social trust. Socialization frequency has a significant positive effect on social mindfulness. It was shown that those with lower socialization frequency had higher levels of social mindfulness. From the value of $R^2$ it is observed that social equity ($R^2 = 0.084$) is better developed than social trust ($R^2 = 0.049$).

Model 7 is for the population born after 1981. Marriage, household registration, pension insurance, home ownership, income level, work status and cultural activities fail to have a significant effect on the social mindset of this group. Gender has only a weakly significant negative effect on social trust. It is manifested in the fact that males have a better level of development than females. There is no effect on social equity. Health insurance has a significant negative effect on social trust. Expressed as a higher level of development for those who have health insurance. No effect on social equity. Physical and mental health have a significant positive impact on social mindfulness. It is shown that the healthier the physical and mental health, the higher the level of social mentality. Housing size has a weakly significant negative effect on social trust. There is no effect on social equity. Economic status has a significant negative effect on social equity. Shows a dominance effect from the top to the bottom. No effect on social trust. Educational attainment has a weakly significant positive effect on social mentality. It is shown that the higher the level of education, the higher the level of social mindfulness.

Socialization frequency has a significant positive effect on social trust. It shows that the lower the frequency of socialization the better the level of development. There is no effect on social equity. From the value of $R^2$, it is observed that social trust ($R^2 = 0.047$) is better developed than social equity ($R^2 = 0.041$).

## The impact of the movement on the social mentality of the Chinese people under spatial differences

The statistical results according to the different spatial divisions are shown in Table 7. This part is still a spatial micro-exploration, and the effectiveness of the campaign on the social mentality of the Chinese people is also low, with only a weakly significant contribution to the social mentality of the people in the central region.

Model 8 is for the population of Eastern China. Household registration, housing ownership, income level, work status and cultural activities failed to significantly influence this group. Gender, age, medical insurance, and economic status had a significant negative effect on social mentality. This is reflected in the fact that males are more likely than females to develop a social mindset; Generations "T, B, X, Y" show a decreasing social mentality, i.e., T > B > X > Y; those who have health insurance have a better social mentality than those who do not; and there is a dominance effect from the top to the bottom in terms of economic status. Pension insurance and mental health have a significant positive effect on social mentality. The level of social mentality is higher among those who do not have pension insurance, and the higher the level of mental health, the higher the level of social mentality. Marriage, housing size, education and socialization frequency have a significant positive effect on social trust. Married is preferred to single; the larger the housing area, the higher the level of social trust; the higher the level of education, the higher the level of social trust; and the lower the

**Table 7. Analysis of the impact of sports on the social mentality of the Chinese people under spatial differences.**

| Variables | Model 8 (Eastern area) | | Model 9 (Middle area) | | Model 10 (Western area) | |
|---|---|---|---|---|---|---|
| | ST | SF | ST | SF | ST | SF |
| Gender | -.103(.038)** | -.075(.036)* | .034(.046) | -.067(.044) | -.112(.049)* | -.092(.048)+ |
| Generation | -.158(.026)*** | -.109(.024)*** | -.166(.030)*** | -.144(.029)*** | -.210(.032)*** | -.165(.031)*** |
| Marriage | .103(.042)* | -.008(.040) | -.081(.051) | .005(.049) | -.069(.054) | -.075(.052) |
| Household Registration | -.016(.042) | -.065(.041) | -.066(.048) | .019(.047) | .032(.056) | -.072(.055) |
| Medical insurance | -.304(.076)*** | -.213(.073)** | .072(.105) | .053(.101) | .014(.105) | -.062(.102) |
| Pension | .120(.048)* | .116(.046)* | .020(.050) | .027(.048) | -.106(.054)* | -.041(.052) |
| Physical health | .010(.020) | .059(.019)** | .035(.023) | .059(.022)** | -.010(.024) | .009(.023) |
| Mental health | .096(.019)*** | .119(.018)*** | .099(.022)*** | .069(.021)** | .088(.023)*** | .117(.022)*** |
| Exercise | -.011(.012) | -.009(.011) | -.027(.014)+ | -.024(.013)+ | -.014(.015) | .015(.015) |
| Housing size | .001(.001)** | -.001(.001) | -.001(.001)+ | -.001(.001) | -.001(.001) | -.001(.001) |
| Homeownership | .031(.038) | .017(.036) | .056(.046) | -.027(.044) | -.029(.049) | -.059(.048) |
| Income levels | -.022(.026) | -.025(.025) | -.040(.029) | -.004(.028) | -.032(.032) | .016(.031) |
| Economic status | -.064(.021)** | -.201(.020)*** | -.095(.024)*** | -.128(.023)*** | -.016(.027) | -.119(.026)*** |
| Employment status | -.038(.044) | .005(.042) | -.005(.047) | -.058(.045) | -.134(.050)** | -.030(.048) |
| Cultural activities | -.024(.023) | -.028(.022) | -.053(.031)+ | -.058(.030)+ | .008(.033) | .001(.032) |
| Educational level | .054(.024)* | -.004(.023) | .014(.031) | .059(.030)+ | .071(.033)* | .005(.032) |
| Social frequency | .048(.016)** | .009(.016) | .023(.019) | .004(.018) | .039(.021)+ | .001(.020) |
| Constant | 3.773(.230)*** | 4.030(.220)*** | 4.301(.268)*** | 4.172(.257)*** | 4.137(.279)*** | 4.065(.270)*** |
| N | 2805 | 2805 | 2139 | 2139 | 1802 | 1802 |
| $R^2$ | 0.058 | 0.089 | 0.050 | 0.054 | 0.053 | 0.057 |

frequency of socialization, the higher the level of social trust. No effect on social equity. Physical health has a significant positive impact on social equity. Expressed as higher levels of individual physical health and higher levels of social equity. There is no effect on social trust. From the value of $R^2$ it is observed that social equity ($R^2 = 0.089$) is better developed than social trust ($R^2 = 0.058$).

Model 9 is for the population of central China. Gender, marriage, household registration, medical insurance, pension insurance, housing ownership, income level, work status and socialization frequency failed to significantly influence the social mentality of this group. Age, economic status, and cultural activities had a significant negative effect on social mentality. The social mentality of the "T, B, X, Y" generations shows a decreasing trend, i.e., T > B > X > Y; the economic status shows a dominant effect from the upper class to the lower class; and the higher the frequency of participation in cultural activities, the higher the level of social mentality. However, cultural activities are weakly significant. Mental health has a significant positive impact on social mindfulness. This is manifested by the fact that the higher the level of mental health, the higher the level of social mentality. Housing size has a weakly significant negative effect on social trust. No effect on social equity. Physical health and education have a significant positive impact on social equity. It is shown that higher levels of physical health and education are associated with higher levels of social equity. However, educational attainment is weakly significant. There is no effect on social trust. From the value of $R^2$ it is observed that social equity ($R^2 = 0.054$) is better developed than social trust ($R^2 = 0.050$).

Model 10 is for the population in western China. Marriage, household registration, medical insurance, physical health, housing size, housing ownership, income level and cultural activities failed to significantly influence the social mentality of this group. Gender and age had a significant negative effect on social mentality. This is reflected in the fact that males have a higher level of development than females, and that the social mentality of the "T, B, X, Y" generations is decreasing, i.e., generation T > generation B > generation X > generation Y. However, gender is weakly significant in social justice. Mental health has a significant positive effect on social mindset. It is shown that the higher the level of mental health, the higher the level of social mentality. Pension insurance and work status have a significant negative effect on social trust. Expressed as higher level of social trust for those who have pension insurance and those who have a worker. No effect on social equity. Education and socialization frequency have a significant positive effect on social trust. It is shown that the higher the education level, the higher the level of social trust, the lower the socialization frequency, the higher the level of social trust. There is no effect on social equity. Economic status has a significant negative effect on social equity. Expressed as a dominance effect from the upper to the lower class. There is no effect on social trust. From the value of $R^2$, it is observed that social equity ($R^2 = 0.057$) is better developed than social trust ($R^2 = 0.053$).

## Deliberations

The topic of " The impact of the exercise on the social mentality of the Chinese people " presents several findings that need to be discussed.

1. **$R^2$ and Adj $R^2$ in the statistical results of this study are not statistically significantly different.** Adj R2 evolved from avoiding statistical distortion of R2 due to a low sample size to variable ratio (i.e., sample size: variable < 5:1) [34]. Of the above models constructed, Model 4 has the lowest sample size (480). The total number of variables chosen for this paper is 20, i.e., 480:20 = 24:1 (still greater than 5:1). This is one of the original reasons for determining that $R^2$ and Adj $R^2$ are not statistically significantly different in this paper. As can be seen through Table 3, among the six statistical results of the three models. The

maximum difference between R² and Adj R² is 0.003, which fails to reach a statistically significant difference [33]. This is the second reason for determining that there is no statistically significant difference between $R^2$ and Adj $R^2$ in this paper. Therefore, Adj $R^2$ is no longer presented after Table 3.

2. **The weighting of social trust and social equity in the social mindset varies.** Across all statistical models, the $R^2$ values for social trust and social equity are inconsistent. This indicates that the two are inconsistent in their ability to explain social mindset. The specific variables that are significant for social trust and social equity in most models are also inconsistent. The values of the significant specific variables and the model explanatory rate, $R^2$, vary according to the inquiry question. And their weight shares fluctuated. This validates the findings of 2 scholars, Pinar [31] (2013) and Qiu [32] (2018), who pointed out that multivariable should be counted separately when they reflect common problems.

3. **The movement has contributed significantly to the social mentality of the Chinese people.** Based on the statistical results, 2 original reasons are visible. One is after adding the exercise variables to the base model. Only the coefficients and standard errors produced mild fluctuations among the variables. No fluctuations in significance, level of significance, or direction of influence occurred and the exercise variable was significant [33]. That is, the movement variables are perfectly adapted to the social mentality of the Chinese people. Secondly, the inclusion of the movement variables caused an increase in the explanatory rate $R^2$ of the model, which improved the degree of explanation of the social mentality [35] (social trust: model 1 $R^2$ = 0.034, model 2 $R^2$ = 0.036. (Social Fairness: Model 1 $R^2$ = 0.038, Model 2 $R^2$ = 0.039). The coding of the sports variable was reversed, which led to the presentation of negative results. That is, the higher the frequency of participation in sports, the better the social mentality of the Chinese people. The positive direction of movement in Model 3 is due to the presence of the interaction variable, which leads to a change in the direction of the variables constituting the interaction term when the model is counted, which is affected by the interaction [34]. However, the movement variable is still significant, indicating that it can still significantly influence the social mentality of the Chinese people.

4. **Differential impact of the movement on the social mentality of the Chinese people in terms of stratification (see model 3).**① There is no impact on the size of housing, home ownership or employment status. As China has implemented the policy of "housing without speculation", the price level of the property market has been stabilized to ensure that the housing needs of the immediate need group can be met on the one hand, and to prevent disorderly growth in housing prices or malicious competition in the property market on the other. It has maintained social equity after effectively regulating the property market and has created a strong social trust among a large group of Chinese people. As a result, the campaign has had no impact on the social mentality of the Chinese people in terms of housing size or home ownership. Entering the new stage of development, the transformation of labor marketization has been accelerating, and the people have changed their self-sufficient way of working to selling their own labor in exchange for remuneration to maintain their livelihood. As a result, the work situation is unable to have a significant impact on the social mentality of the Chinese people.② Income level, education level and socialization frequency have a significant contribution to social trust, with no effect on social equity. The high-quality development of China's economy promotes social trust by lowering people's income risk, especially for low-income groups, and this economic risk barrier strengthens the economic strength of individuals or families and promotes their social trust. As financial inclusion and other related policies remain at the macro-planning level and are only

effectively implemented in some cities, banks still have differentiated services for different classes, making it difficult to put people's financial well-being into practice, and resulting in an income hierarchy that fails to significantly contribute to people's social equity. To enhance people's knowledge and education, China introduced compulsory education in 1986, and since then has basically realized equity in basic education, so that educational attainment has not been able to influence social equity. However, the continuous improvement of the people's knowledge has contributed to the continuous strengthening of social trust. The increase in socialization frequency facilitates people's multi-perspective and all-round understanding of society and promotes social trust. From the existing studies, there is no research that confirms the existence of a significant mutual influence relationship between socialization frequency and social equity.③ Economic status contributes significantly to the social mentality of the Chinese people, with a decreasing effect from the top to the bottom. The same suggests that economic competence, economic risk barriers, and high-quality economic development are important ways to enhance the social mindset of the Chinese people.④ Cultural activities have a weak and significant contribution to social equity. Participating in more cultural activities is conducive to inculcating socialist core values and promoting social equity through the dissemination and learning of cultural knowledge [34].

5. **Why are movement variables significant at the macro level but have no significant effect at the micro level?** Models 2 and 3 in Table 3 provide a macro-level exploration of the movement's impact on the overall social mindset of the Chinese people. The results present a strong significance. In Tables 6 and 7, the micro-levels of intergenerational and spatial inquiry are presented. The results show that the movement failed to influence the social mindset of the Chinese people significantly or weakly significantly. Why are the results of the same data source inconsistent at the macro-level and micro-level? This is due to the problem of the depth of the social mentality of the Chinese people that the movement can influence [34]. The significance of the interaction terms between exercise and generation, and exercise and space, then confirms that exercise produces significant effects on generation and space [33]. The current movement can affect only the most macroscopic social state of mind of the Chinese people. That is, it is the most macroscopic socio-psychological relationship as pointed out by the Chinese scholar Yang Yiyin [5]. Its dimension of influence and its ability to influence are mainly exerted at the macro level. The micro level is not significant or weakly significant due to its current lack of impact capacity. This does not mean that it will not have a significant impact in the future. However, it requires the formulation of policies that fit the context of the times and the use of precise methods and tools to achieve this. Eventually, the campaign will be able to guide the social mentality of the Chinese people in a significantly positive way at both the macro and micro levels.

6. **An overview of the statistical results of all the models shows why there is a paradoxical situation where many variables are significant but the models have a low rate of explanation.** From basic models with only control variables, to movement models with core effects, to hierarchical models, and models with sub generational and spatial discussions. Except for the social trust orientation in Model 5, the explanatory rates of all the models are slightly higher than that of Model 1. At the same time, each model has a different number of variables significant among them. However, the overall observation shows a low value of $R^2$. This is since there are numerous variables that can significantly influence the social mindset of the Chinese people and each variable can also provide a role. However, in real life the factors failed to meet the needs of the people [35]. It leads to a situation where numerous variables are significant, but the model has a low rate of explanation [36]. This

suggests that there is enormous room for improvement in the social mindset of the Chinese people, which has been enhanced by the Movement. Therefore, there is an urgent need to clarify the specific circumstances affecting the social mentality of the Chinese people. Targeted measures should be taken to increase the implementation of various aspects to meet the needs of the people and thus improve the social mindset affected by the movement.

7. **Digging deeper into class differences to optimize the mechanism of the movement's influence on social mentality.** This can be seen through Model 3 in Table 3. The class differences in the impact of the movement on the social mentality of the Chinese people are obvious and complex. To optimize the mechanism of the movement's impact on the social mentality of the people, it is necessary to dig deeper into the class differences involved. The first and foremost thing to be upgraded is the economic status of the people. This is both because the statistical results show a strong significance effect and because the social mentality is reflected through the people's eagerness to improve their economic status. Fittingly, Ohl (2000) scholars concluded that Bourdieu's social class theory plays an important role in explaining human behavior [37]. The next thing that needs to be upgraded is the income level of the people. Strengthening the implementation of financial inclusion and other related policies, prohibiting banks from treating customers differently according to their class, and establishing a fair social trust orientation. Optimize social frequency. Currently, telecommunication fraud seriously affects social trust, and there are many different methods of telecommunication fraud, which make the victims unable to defend themselves. There is a choice to reduce the social frequency to avoid losses. This leads to the fact that the lower the social frequency, the higher the level of social trust [38]. To this end, it can be done by cutting off the intermediary means of telecommunication fraud through operators, strengthening the auditing efforts of banks on abnormally large remittances, and vigorously publicizing the means and methods used in telecommunication fraud, so that avoiding being defrauded is an important way to enhance the social mentality of the people.

8. **Balancing intergenerational differences and building the foundation of a social mindset for movement improvement.** When analyzed by generation, the exercise variable, which is the core of the impact, was weakly significant or not significant. However, under the influence of exercise, the remaining variables showed unique significant effects depending on the generation. Social trust in the social mentality of the movement-affected Chinese people is manifested in: Generation "T" ($R^2 = 0.085$) has the best level of development. Generation "X" ($R^2 = 0.049$) has the next highest level of development. Generation "Y" ($R^2 = 0.047$) has the 3rd highest level of development. Generation "B" ($R^2 = 0.023$) had the worst level of development. Therefore, in social trust, emphasis should be placed on increasing the level of development of social trust in generation "B" and stimulating the level of social trust in generations "X" and "Y". The social justice in the social mentality of the Chinese people influenced by the movement is manifested according to different generations: Generation "X" ($R^2 = 0.084$) has the best level of development. Generation "T" ($R^2 = 0.067$) has the next highest level of development. Generation "B" ($R^2 = 0.055$) has the 3rd highest level of development. Generation "Y" ($R^2 = 0.041$) had the worst level of development. Therefore, in terms of social equity, emphasis should be placed on increasing the level of development of the social mentality of generation "Y" and strengthening the level of social equity of generations "T" and "B". The above data confirms that the consideration of generational differences in policy making is an important reflection of a country's ability to govern [39]. The various results derived from this paper can also be used as a reference point for the relevant authorities when formulating policies. With exercise as the core of influence, it positively guided the social mindset of the people, while mental health and physical health were

significant. Indirectly, it is confirmed that health prevention and enhancement systems can significantly influence human longevity and improve social mindset in different generational groups [21].

9. **Capturing spatial characteristics to energize the space in which the movement influences the social mindset.** This study confirms that there are spatial differences in the effects of exercise on the social mindset of the Chinese people. However, little has been done to explore how exercise enhances people's social mindset under spatial conditions. The effect of exercise on social trust was demonstrated under spatial conditions: The East ($R^2 = 0.089$) has the best level of development, the West ($R^2 = 0.057$) has the next best level of development, and the Center ($R^2 = 0.054$) has the worst level of development, but it is comparable to the West. Social fairness is manifested in: The East ($R^2 = 0.058$) has the best level of development, the West ($R^2 = 0.053$) has the next best level of development, and the Center ($R^2 = 0.050$) has the worst level of development, but the overall difference is not significant. Therefore, the characteristics of different spaces must be specifically analyzed to improve the social mindset in general. Of these, age (generation), mental health and economic status were able to significantly influence residents in all 3 spaces. To this end, these three factors should be closely scrutinized, focusing on improving the socio-mental development of young people in order to balance the age (generational) gap; developing the level of mental health of the people, creating a strong inner world and strengthening the psychological foundation; and reducing the differences in economic status, which, as can be seen in Table 7, is significant, but not necessarily at the level of income. Studies by Negacz (2021) [27] and Vicari (2013) [36] point out that this is since economic status can be a means of generating both income for oneself and the well-being of the people, and a means of transcending status hierarchies. Whereas an increase in income levels does not necessarily lead to an increase in economic status. For example, the results of scientists and researchers can be transformed into economic benefits, which not only benefit themselves, but also contribute to the development of society, increase the income of the people, and gain higher honor and prestige. Then there are the relevant policies, systems and rules formulated by politicians, entrepreneurs, etc., and the implementation of certain project works as well as development strategies, which contribute to the development of the country and improve the country's governance capacity and comprehensive national strength, while benefiting the people at the same time. On the contrary, an increase in income level alone may not promote national development, social development, and human progress [37]. Therefore, balancing age (generational) differences and differences in economic status to strengthen mental health should also be approached according to spatial characteristics. This is complemented by factors specific to the characteristics of different spaces to stimulate the role of sports in influencing people's social mentality.

## Conclusion

As we enter the 21st century, the increasing productivity has prompted the Chinese people to diversify their needs. There is a gradual shift from the initial goal of achieving a long and healthy life through the natural attributes of exercise to a reliance on the social attributes of exercise. It presents a broad, macroscopic, individual and group social mentality, and is an important way of reflecting the will of the people. Enhancing the social mindset of the Chinese people through sports will help fulfill the people's quest for a long and healthy life and promote the continuous improvement of China's level of human development. To this end, this paper constructs a statistical model using CGSS data released in 2023, considering considerations of

class, generation, and space. It is confirmed that sports have a significant contributing effect on the social mentality of the Chinese people. This is manifested in the fact that the higher the frequency of participation in sports, the higher the level of development of social mentality. At the same time, a situation of class differences is presented. Economic status has a significant contribution to social mentality. The benefit of the effect is characterized by a decreasing effect from the top to the bottom. Income level, education, and socialization frequency have a significant contribution to social trust and no effect on social equity. This is manifested as a decreasing effect of income level from high to low, an increasing effect of education level from low to high, and a decreasing effect of socialization frequency. Cultural activities have a weak and significant contribution to social equity. This is reflected in the fact that the higher the frequency of participation in cultural activities, the higher the development of social equity. As evidenced by the results of the interaction item test, there are generational and spatial differential effects of exercise on the social mindset of the Chinese people. Under different generational and spatial conditions, the impact of the movement on the social mentality of the Chinese people varies. Further analysis reveals that this effect is significant only at the macro level, but not significant or weakly significant at the micro level (by generation and by space). However, the effectiveness of the influence of class permeates over different generational and spatial levels. It is indicated that the effectiveness of the current movement on social mentality is limited and has not yet met the needs of the people, while alleviating class differences can enhance the social mentality of the people. To this end, policy instruments must be developed to stimulate the enthusiasm and participation of people of different generations and spaces in the movement. Enhance the positive effects of sports on their social mentality, while utilizing the influence provided by class. Looking at all the statistical models, the $R^2$ values of the 2 dimensions of social trust and social fairness under social mindset did not show consistency. It indicates that the social mentality of the Chinese people influenced by the movement presents the internal dimensions with different weight shares. And the weighting percentage fluctuates with the inquiry question. Also, there is no statistically significant difference between $R^2$ and Adj $R^2$. For this reason, it is recommended that when exploring the topic of social mentality, the internal dimensions of social trust and social fairness should be explored separately, to ensure the stability and reliability of the reflection of social mentality, which is more conducive to the further refinement of the research issues. It is recommended to dig deep into class differences to optimize the mechanism of the movement's influence on social mentality; to balance intergenerational differences to build a solid foundation for the movement to improve social mentality; and to grasp spatial characteristics to stimulate the spatial vitality of the movement's influence on social mentality.

## Supporting information

**S1 Data. Chinese general social survey data, CGSS.**
(DTA)

## Author Contributions

**Conceptualization:** Shuyu Ji, Xiannan Yang.

**Data curation:** Shuyu Ji, Kaiqi Zhang.

**Formal analysis:** Shuyu Ji, Kaiqi Zhang.

**Methodology:** Shuyu Ji, Xiannan Yang.

**Software:** Shuyu Ji, Kaiqi Zhang.

**Supervision:** Xiannan Yang.

**Validation:** Shuyu Ji.

**Writing – original draft:** Shuyu Ji, Ludan Xu.

**Writing – review & editing:** Xiaolin Wang, Delong Dong, Xiannan Yang.

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
