## [Decision Letter · Decision Letter 0]

8 Apr 2024

PONE-D-23-36514Social Class, Generational Disparities, and Spatial Dimensions: How Does Exercise Impact the Social Mentality of the Chinese Population?PLOS ONE

Dear Dr. Dong,

Thank you for submitting your manuscript to PLOS ONE. After careful consideration, we feel that it has merit but does not fully meet PLOS ONE’s publication criteria as it currently stands. Therefore, we invite you to submit a revised version of the manuscript that addresses the points raised during the review process.

We look forward to receiving your revised manuscript.

Kind regards,

Robbert Huijsman, PhD

Academic Editor

PLOS ONE

Journal Requirements:

Additional Editor Comments:

Although there is only one reviewer's report, we agree with that review. To avoid any further delay, we decide on major revision and request you to follow up on the reviewer's comments and advises.

Reviewers' comments:

Reviewer's Responses to Questions

**Comments to the Author**

1. Is the manuscript technically sound, and do the data support the conclusions?

Reviewer #1: Partly

2. Has the statistical analysis been performed appropriately and rigorously? 

Reviewer #1: No

3. Have the authors made all data underlying the findings in their manuscript fully available?

Reviewer #1: No

4. Is the manuscript presented in an intelligible fashion and written in standard English?

Reviewer #1: No

5. Review Comments to the Author

Reviewer #1: Journal: PLOS ONE

Article title: Social Class, Generational Disparities, and Spatial Dimensions: How Does Exercise Impact the Social Mentality of the Chinese Population?

Manuscript ID: PONE-D-23-36514

General Comments:

This article studies the influence of exercise, specifically through the lens of sports, on the social mentality of the Chinese population, encompassing aspects of social trust and social equity with the distinct mechanisms underlying differences in social class, generation disparities, and spatial dimensions. The authors used the data from the 2023 China General Social Survey, by selecting 20 variables and analyzing a sample of 6,746 individuals, with ordinary least squares (OLS) multiple linear regression models. The authors reached the conclusions that exercise has a positive influence on the social mentality of the Chinese population; a higher frequency of participation in sports correlates with a more advanced level of social mentality development; social class, generational disparities, and spatial dimensions demonstrate substantial impact, each exhibiting unique characteristics depending on the specific research question.

Overview:

The paper is good written and the empirical work does appear to be carefully and correctly done. The research question is very good and it does make a sufficient new contribution to the literature to be suitable for the PLOS ONE ONLY after MINOR revisons.

In fact, the literature on the social class, generational disparities, and spatial dimensions: how does exercise impact the social mentality of the Chinese population is quite new.

The contribution of the paper is the analysis of 2023 China General Social Survey, by selecting 20 variables and analyzing a sample of 6,746 individuals, with ordinary least squares (OLS) multiple linear regression models.

The paper is very interesting; and in my view, it needs to be MAJOR improved to reach the standard required for publication in this journal.

Specific Comments:

1. Title: quite long; try to reduce to one sentence

2. Abstract: good

3. Introduction: NOVELTY + results (better explanation);

4. Literature review: for the part of social class theory, the authors must introduce newer literature (and not only Chinese). This part is very political and subjective.

5. Methodology: why the authors use only these indicators into the model? Present some theoretical explanations for these indicators

6. Model: Why in model 3/4/5, the authors introduce the variable class, which is very subjective and not used into analysis after that?

7. Model: the authors must introduce some tests for multicollinearity and endogeneity.

8. Model: the authors must introduce a robustness part.

9. Conclusions: at least 1 page

General considerations: the idea of the article is somehow mediocre, but the construction of the article is sometimes very technical (statistical) and political. The authors MUST improve the methodology, explanations, and change the article accordingly. The authors MUST remove the political and subjective part of the article.

I ONLY recommend this article be published in PLOS ONE after MAJOR revisions (models and the social class theory).

6. PLOS authors have the option to publish the peer review history of their article (what does this mean?). If published, this will include your full peer review and any attached files.

Reviewer #1: No

---

## [Author Response · Author response to Decision Letter 0]

20 May 2024

Dear Reviewers:

 Greetings! Thank you very much for your valuable comments and suggestions on this article, we have benefited a lot.For each comment and suggestion, we have made careful deliberation and consideration, and carefully reviewed the relevant literature and information, and made corresponding modifications, and now the specific modifications are fed back as follows:

I、Responses to modifications

1. Title: quite long; try to reduce to one sentence

The author is very much in favor of your opinion. The title has been changed to "The impact of the exercise on the social mentality of the Chinese people". (line 1).

2. Abstract: good

No changes have been made here.

3. Introduction: NOVELTY + results (better explanation)

The author strongly agrees with your comments. Therefore, the introduction and literature review sections have been blended. The ideas presented in different studies are sorted out to provide arguments for this study. The argument is developed from the shift in people's need for sport from natural to social attributes, the positive and negative states presented by people's social mentality, and the impact of sport on social mentality. The effects and differences brought about by class, generation and space are also described. (lines 45-114).

4. Literature review: for the part of social class theory, the authors must introduce newer literature (and not only Chinese). This part is very political and subjective.

This has been integrated with the introductory section. Based on your comments, new literature has been introduced. The discussion of class is developed from different perspectives. Specifically: the relationship between people's trust in government and class; the dominance relationship between classes and social mentality; the impact of class-differentiated services of financial institutions on social mentality; and the relationship between the demand for sport and class mobility. (lines 76-92).

5. Methodology: why the authors use only these indicators into the model? Present some theoretical explanations for these indicators

The theoretical explanations for the selection of model indicators have been further expanded. See the indicator selection section in the research design. (lines 127-155).

6. Model: Why in model 3/4/5, the authors introduce the variable class, which is very subjective and not used into analysis after that?

The author is in deep agreement with you on this comment. The explanation of the model is enhanced with modifications. Located in the explanation and discussion of the statistical results, respectively. (lines 262-412, and lines 445-473).

7. Model: the authors must introduce some tests for multicollinearity and endogeneity.

We have gained a lot from this comment of yours. It motivates us to further standardize the use of statistical models and also further increase the reliability of the statistical results. Multicollinearity diagnostics added to the model setup section. Endogeneity tests were added to the statistical results section. Endogeneity tests were completed using the instrumental variables method. The original motion variable was replaced with an instrumental variable for whether or not the person participated in the motion. (lines 165-171, and lines 281-289).

8. Model: the authors must introduce a robustness part.

We have benefited greatly from this observation of yours. Accordingly, we used 2 methods for robustness testing, replacing the dependent variable and propensity score matching. It has been added in the statistical results section. The combination of social trust and social fairness as social mindfulness was used as a replacement for the dependent variable. Propensity score matching was performed to divide the experimental and control groups with the mean of the exercise variable, and the operation was completed using the control variable as the matching variable. (lines 290-297).

9. Conclusions: at least 1 page

Based on this comment of yours, we have expanded our conclusions. (lines 568-602).

II、Explanation of other changes

1、Some of the discussion in the text has been reorganized and corrected, and references and their serial numbers have been added and updated.

2、Reviewing the whole text, certain information details and language expressions in the text have been revised to make the presentation of views more accurate and concise.

III、 the revised parts of the article have been marked in red font, please review them again!

Finally, once again, I would like to express my heartfelt thanks to you for your valuable revisions and suggestions! I wish you happiness, well-being and success in your work!

---

## [Editor Report · Decision Letter 1]

10 Jun 2024

The impact of the exercise on the social mentality of the Chinese people

PONE-D-23-36514R1

Dear Dr. Dong,

We’re pleased to inform you that your manuscript has been judged scientifically suitable for publication and will be formally accepted for publication once it meets all outstanding technical requirements.

Kind regards,

Robbert Huijsman, PhD

Academic Editor

PLOS ONE

---

## [Editor Report · Acceptance letter]

26 Jun 2024

PONE-D-23-36514R1 

PLOS ONE

Dear Dr. Dong, 

I'm pleased to inform you that your manuscript has been deemed suitable for publication in PLOS ONE. Congratulations! Your manuscript is now being handed over to our production team.

Kind regards, 

on behalf of

Professor Robbert Huijsman 

Academic Editor

PLOS ONE